# Controlled Mandibular Repositioning: A Novel Approach for Treatment of TMDs

**DOI:** 10.3390/bioengineering12080865

**Published:** 2025-08-11

**Authors:** Diwakar Singh, Alain Landry, Martina Schmid-Schwap, Eva Piehslinger, André Gahleitner, Thomas Holzinger, Yilin Wang, Jiang Chen, Xiaohui Rausch-Fan

**Affiliations:** 1Center for Clinical Research, University Clinic of Dentistry, Medical University of Vienna, 1090 Vienna, Austria or dentidtdiwakarsingh@gmail.com (D.S.); thomas.holzinger@meduniwien.ac.at (T.H.); n12046947@students.meduniwien.ac.at (Y.W.); 2Department of Education in Occlusion Medicine, Vienna School of Interdisciplinary Dentistry (VieSID), Advanced Education, 3400 Klosterneuburg, Austria; i.d.alainlandry@gmail.com; 3Division of Prosthodontics, University Clinic of Dentistry, Medical University of Vienna, 1090 Vienna, Austria; martina.schmid-schwap@meduniwien.ac.at (M.S.-S.); eva.piehslinger@meduniwien.ac.at (E.P.); 4Division of Radiology, University Clinic of Dentistry, Medical University of Vienna, 1090 Vienna, Austria; andre.gahleitner@meduniwien.ac.at; 5Competence Center Artificial Intelligence in Dentistry, University Clinic of Dentistry, Medical University of Vienna, 1090 Vienna, Austria; 6Comprehensive Center for Artificial Intelligence in Medicine, Medical University of Vienna, 1090 Vienna, Austria; 7School and Hospital of Stomatology, Fujian Medical University, Fuzhou 350005, China; dentistjiang@sina.com

**Keywords:** controlled mandibular repositioning, anterior repositioning splint, temporomandibular joint disorders, condylar position variator, condylography, therapeutic position, disc displacement with reduction, controlled mandibular repositioning stabilizer

## Abstract

Temporomandibular joint disorders (TMDs), particularly disc displacement with reduction (DDwR), are prevalent musculoskeletal conditions characterized by symptoms such as joint clicking, pain, and sometimes limited jaw movements. Accurate diagnosis requires a multidisciplinary approach, including clinical examination, imaging (MRI), and functional analysis. Among conservative treatment modalities, anterior repositioning splints (ARSs) are widely used to recapture the displaced discs and reposition the mandibular condyles. Determining the optimal therapeutic position (Th.P) for anterior repositioning splint fabrication remains challenging due to individual anatomical variability and a lack of standardized guidelines. This study introduces the controlled mandibular repositioning (CMR) method, which integrates clinical examination, imaging (MRI), computerized cephalometry, computerized condylography, neuromuscular palpation, and the Condylar Position Variator (CPV) to define an individualized Th.P. After treatment with CMR stabilizers (splints), the control MRI confirmed that in 36 out of 37 joints, the discs were repositioned to their normal position. There was a reduction in pain, as shown by VAS scores at the 6-month follow-up. This study demonstrated the effectiveness of the CMR method to find a precise therapeutic position, resulting in a 97.3% joint luxation reduction in DDwR. This study underscores the importance of precise, individualized Th.P determination for effective anterior repositioning.

## 1. Introduction

The DC-TMD consortium recommends history and examination, clinical diagnosis, and imaging (MRI: magnetic resonance imaging) for the diagnosis of intra-articular joint disorders [1]. The DC-TMD criteria classify these intra-articular joint disorders as disc displacement with reduction, disc displacement with reduction with intermittent locking, disc displacement without reduction with limited opening, or disc displacement without reduction without limited opening [1]. Based on a meta-analysis, the reported global incidence of TMDs is 34%, and it varies between different continents: (Asia—33%, South America—47%, North America—26%, Europe—29%) [2]. A few studies have reported the prevalence of temporomandibular joint disorders (TMDs), which varies from 20 to 60% in children and adolescents [2,3]. The most commonly reported TMD symptoms were clicking sounds and pain. The prevalence of TMJ clicking ranges from 18 to 35% [4,5,6].

The treatment of TMDs varies from conservative to multimodal, which includes behavioural therapy, stress management, physiotherapy, pharmacotherapy, moist heat application, needling techniques, occlusal devices (occlusal appliances, splints), and minimally invasive procedures, which include arthrocentesis and intra-articular injections [7]. Depending upon the underlying joint pathology, conservative therapy up to open joint surgeries are also recommended [7,8]. From the conservative treatment approach for treatment of TMDs, occlusal appliances, occlusal splints, or orthotics appliances are the common choice for dental practitioners [9,10]. Among the wide variety of splints, anterior repositioning splints (ARSs) are the method of choice and are commonly suggested for DDwR [11,12,13]. ARSs allow the mandibular condyle to be repositioned anteriorly in order to recapture the displaced disc. This anterior repositioning also helps to reduce joint sounds and pain [14].

The range of mandibular repositioning varies in previous studies, showing the following variations: (I) a minimal amount of anterior repositioning; (II) up to 2 mm of anterior repositioning; (III) up to 3 mm of anterior repositioning; (IV) the most common position is achieved by going protrusive to an edge-to-edge incisors position, where the joint luxation can be clinically reduced and considered a therapeutic position for anterior repositioning splints [15,16,17]. For cases where the anterior repositioning is suspected to be too far anteriorly, during the follow-up appointments, subsequent grinding of the splints is performed in order to ‘walk back’ the condyles [16], and sometimes this is known as step-back anterior repositioning [18]. As described above, the proposed anterior repositioning is carried out clinically based on individual experience [15,16,17].

In the literature, there is an uncertainty in the programming of articulators and the methods used to calculate the sagittal condylar inclination (SCI) through protrusive occlusal records, OPGs (Orthopantomograms), and CBCT (Cone Beam Computed Tomography) [19]. The average condylar inclination measured by axiography is 42.1°, as compared to that of interocclusal records, which is 33.25° [20]. In one of the studies, the authors set the SCI at 30° to program the articulators for the fabrication of ARSs [21]. The sagittal condylar inclination refers to the angle formed between the horizontal plane (axis–orbital plane) and the condylar tracing inclination during protrusive movement over the posterior slope of the articular eminence. This sagittal condylar inclination helps to determine the movement of the mandible during various functions, such as mastication, speech, swallowing, and excentric movements [22]. Discrepancy in this sagittal condylar inclination may result in unnecessary adjustments on occlusion [19].

In order to fabricate occlusal appliances or anterior repositioning splints, or even prosthetic rehabilitation, studies have recommended proper sagittal condylar inclination values [19]. One of the conclusions from the latest systematic review mentioned that digital axiography/condylography has a higher potential to capture comprehensive functional data and may surpass other techniques [19]. The computerized condylography is precise and very well studied [23,24,25,26,27,28].

The CADIAX (computerized axiography/condylography) is considered as a gold standard for functional diagnostics and various studies have been performed to evaluate the reproducibility, comparison with other jaw tracking devices and its clinical implications [23,29,30,31]. A few studies have been performed based on instrumental functional analysis to calculate the therapeutic position [32,33,34].

The aim of this study is to calculate therapeutic position by means of the controlled mandibular repositioning (C.M.R.) method following a thorough clinical examination and diagnosis. It is based on an individual computer-aided functional analysis (for computerized cephalometry and condylography) in patients with DDwR.

The therapeutic position can be calculated either by the angular method or by the coordinate method during controlled mandibular repositioning. The CMR method takes into account the reaction of the craniomandibular system and the neuromuscular system by using muscle, joint, and ligament palpation.

## 2. Materials and Methods

### 2.1. Selection of Patients

Subjects with disc displacement with reduction (verified with magnetic resonance imaging) were recruited from the University Clinic of Dentistry, Clinical Division of Prosthodontics, Special Clinic for Temporomandibular Disorders, Medical University of Vienna. Ethics approval was obtained from the Medical University of Vienna, Austria (EKNr: 2267/2018). A required sample size of 16 subjects was calculated, assuming an average difference in disc position before and after CMR therapy of 0.5 mm ± 0.5 mm. The calculation was conducted for a paired samples *t*-test assuming a statistical power of 95% and a significance level of 5% (two-sided) to compare the results within the same subjects before and after the treatment To compensate for potential dropouts in the course of the study, 20% (=16 × 0.2 = 3.2 ≈ 4) more subjects were included. The total sample size was, therefore, set to 20 participants. The sample description is shown in Table 1.

Inclusion criteria: Subjects/patients were enrolled based on the following inclusion criteria: (a) age between 18 and 45 years, (b) absence of any systemic diseases, (c) clinical diagnosis of ADDwR based on DC/TMD [1], and (d) MRI confirmation of ADDwR, in at least one of the joints (e) without moderate-to-severe arthrotic changes, and (f) with signed informed consent.

Exclusion criteria: Subjects (a) who were pregnant, (b) with congenital abnormalities or dentofacial deformities, (c) who recently underwent oro-facial surgery, cervical trauma, or history of major accident, (d) with major psychological disorders, (e) with complete or partial dentures, (f) who had received prior TMD treatment, (g) who have two or more teeth missing in one quadrant, (h) subjects with periodontal problems, and (i) subjects with claustrophobia were excluded.

Out of the 20 subjects, 17 patients presented bilateral ADDwR, and the remaining 3 patients presented with ADDwR on one side and ADDw/oR on the other side. All the subjects enrolled in the study had to undergo the standard anamnesis, which included detailed records of each subject’s medical and dental history, the INFORM DC-TMD (https://inform-iadr.com/ (accessed on 29 February 2024)) protocol evaluation of joint clicking, acquisition of dental impression and the fabrication of split dental casts, TMJ MRI, palpation of muscles and TMJ structures, visual analogue scale, and condylography. (This information is from the first paper on MRI part of CMR method published in 2024) [33].

### 2.2. Condylography: CADIAX^®^ 4 (Computerized Condylography) (Gamma-Dental Klosterneuburg, Austria)

Condylography by CADIAX 4 is a digital axiography method for monitoring condylar motion in 3D and is a part of clinical instrumental analysis for functional diagnosis (S2K guidelines German Society of Craniomandibular Function and Disorders https://www.quintessence-publishing.com/downloads/cmf_2023_03_s2k_guideline.pdf (accessed on 10 January 2024) (DGFDT) using CADIAX 4 diagnostics (Gamma dental Klosterneuburg, Austria, https://www.gammadental.com/en/, accessed on 10 January 2024). The high-precision electronics (16 bit ADC) work with an internal measurement resolution of 0.001 mm (https://www.gammadental.com/en/jawtracking.html#slide1, accessed on 10 January 2024).

The system records practically any number of mandibular movements, up to a duration of 18 s per registration. The registration of the condylar movements is performed close to the joints and on the exact individual hinge axis (the True Hinge Axis of each condyle). CADIAX provides all relevant jaw-related motion data for the diagnosis and functional therapy of the occlusion [33].

The condylography records were performed before and after CMR therapy.

CADIAX^®^ 4 measures translation and rotation of each condyle in the three planes of space (X, Y, Z) in reference to the axis–orbital plane. The *X*-axis represents the antero-posterior axis, while the *Z* axis represents the vertical axis and the *Y* axis represents the transversal axis. The starting point of all movements is from the reference position (RP). The RP is the retral border position of the mandible, in which the joint structures are not stressed, according to Rudolf Slavicek, The Masticatory Organ, 2002 [35].

Figure 1a shows a condylograph mounted on a dental manikin along with a kinematic facebow. Figure 1b shows the mounting of the upper dental cast with the kinematic facebow in the True Hinge Axis. Figure 1c shows the upper dental cast transferred to the CPV. The mandibular dental cast is articulated, in the reference position, on the CPV as shown in (Figure 1c), before the controlled mandibular repositioning method is used.

### 2.3. Controlled Mandibular Repositioning Method: Condylar Position Variator (CPV) (Gamma-Dental Klosterneuburg, Austria)

The superior part of the CPV (Figure 1c) has a central part where the upper model is attached. Instead of having condylar housings like those on an articulator, there are sliding blocks that are attached to the central part and aligned with the True Hinge Axis when all the settings are set to zero. These sliding blocks, when the Sagittal Condylar Inclination (S.C.I.) is set at zero, allow pure horizontal movement (*X* axis) and pure vertical movement (*Z* axis). A vernier scale on each axis allows adjustments to be made by 0.1 mm increments. The scale on *X* and *Z* axis goes from −5 to +5 mm. Using the CPV this way is referred to as the coordinate method.

The CPV offers the possibility to adjust the SCI° angle as one would on an articulator, with a range from 0° to 70°. One can use the CPV by adjusting the S.C.I. according to condylographic tracings. Since the *X* axis is rotated at the S.C.I. setting, it is possible to adjust the *X* axis (now angulated at the proper S.C.I.) and to adjust the sliding block to the desired amount of mandibular repositioning.

On the gamma-dental software, this amount is given by the S data. Using the CPV this way was named the angular method. On the right side of the CPV (Figure 1c), there is a short extension which represents the *Y* axis and it is used for transversal axis correction, if needed. On the *Y* axis, the scale goes from −3 to +3 mm.

### 2.4. Controlled Mandibular Repositioning Method

First, the vertical dimension of occlusion (V.D.O.) is evaluated from the digitized cephalometric X-Ray. For example, in the sample subject (Figure 2), the cephalometric data shows a lower facial height of 43.5°. The green colour represents a vertical dimension close to the normal value of 44.4°. In order to have a minimum thickness of 2 mm in the molar area for the fabrication of a CMR stabilizer/splint, the incisal pin height will be increased by +4 mm. If a change to the skeletal vertical dimension is needed, depending upon the skeletal morphology (brachyfacial or dolichofacial) of the particular subject, the incisal pin of the CPV will be adjusted accordingly.

In brachyfacial patients, the vertical dimension will be normalized and will be incorporated into the CMR stabilizer.

For dolichofacial subjects, the vertical dimension is kept at its minimum, but we need 2.0 mm of posterior space between the molars to avoid fracture of the CMR stabilizer.

The Condylar Position Variator (CPV) allows S.C.I. adjustments from 0° up to 70 degrees. In order to reach our therapeutic position, depending on the Sagittal Inclination of the condylar translation (Figure 3a), we have the opportunity to use two different methods: the angular method and the coordinate method.

Angular method: If the SCI at an individual therapeutic position is less than 70° then it is simple to adjust the S.C.I. on the CPV by simply rotating the sliding blocks at the exact S.C.I° given by the Gamma software (Figure 3a). The software calculates the distance (S) from R.P. to the desired Th.P. with a precision of 0.1 mm. ‘S’ shows the amount of anterior repositioning on the *X* axis, which is already inclined to the proper S.C.I. as shown in (Figure 3b).

Coordinate method: If the SCI of the calculated therapeutic position is greater than 70° (Figure 4a) then we use the coordinate method in which the SCI is kept at 0°. We then use the X (antero-posterior) and Z (cranial–caudal) coordinates provided by the Gamma software to adjust the condylar parts (sliding blocks) of the CPV. (Figure 4b depicts the calculations used in the coordinate method).

Both methods can be used to reach the desired therapeutic position point on a condylographic tracing (Figure 4a). Usually, the angular method is used when the S.C.I. is smaller than 70°. The coordinate method is used when the S.C.I. is greater than 70°.

Figure 5a shows the initial MRI conducted before the treatment, where the disc is displaced anteriorly and the condyle is displaced posteriorly. After 6 months of controlled mandibular repositioning stabilizer therapy, the control MRI (Figure 5b) showed the recapturing of the displaced disc.

Controlled Mandibular Repositioning stabilizer: an anterior view is shown in (Figure 6a) and an occlusal view in (Figure 6b). The subjects were advised to wear their CMR stabilizers for more than 20 h a day and only to remove them for eating or when practicing oral hygiene.

The diagram (Figure 7) summarizes the different steps that the subjects went through during the CMR therapy and during the re-evaluation process after 6 months of CMR therapy.

## 3. Results

Statistical analyses were conducted using IBM^®^ SPSS^®^ Statistics, version 29, and R Statistics, version 4.2.1. The MRI evaluation and sensitivity to muscle palpation were described in a publication in 2024, which we denoted as part 1 (this part is from the first paper on the MRI part of the CMR method published in 2024) [33]. In this study, the normality of the paired differences between pre- and post-treatment measurements was assessed by the Shapiro–Wilk test. The Shapiro–Wilk test was chosen because it can adequately handle small sample sizes (*n* < 50) [https://shorturl.at/d0wZa, accessed on 26 July 2025]. Normality was assumed if this test showed type 1 error greater than 0.05 as a quantitative method, and also performed qualitative evaluations.

A significance level of *p* < 0.05 was used for all statistical tests. When the assumption of normality was met, a paired *t*-test was performed; otherwise, the Wilcoxon signed-rank test was applied.

To compare the SCI and the disc displacement pre- and postoperatively, a paired *t*-test was applied to evaluate the effect of the treatment. In cases where normality could not be assumed, the adequative non-parametric test known as the Wilcoxon signed-rank test was used instead.

Spearman rank correlation: In order to quantify the correlation between the angular and the coordinate method, the Spearman rank correlation was applied. This was performed for all joints.

### 3.1. MRI Evaluation of Therapeutic Position (Figure 8)

As explained in paper 1, the MRI evaluation was performed before and after the treatment [33]. In this study, a non-linear measurement was performed based on a study by Q. Zhang et al. [36]. The landmarks were the posterior part of the disc (D) and the superior part of the condyle (C). (L is the line through the condylar axis). The distance between C and D was measured before (Figure 8a) and after treatment (Figure 8b), corresponding to the amount of disc displacement in relation to the condylar head. (Figure 8a) and (Figure 8b) are magnifications of (Figure 5a) and (Figure 5b) only.

Figure 8c shows the distance between the condylar axis and the posterior band of the disc in millimetres, before and after CMR therapy.

There was a statistically significant difference with respect to disc position in mm after CMR therapy compared to before for right *t* (18) = −13.533, *p* < 0.001) and for left joint *t* (19) = −16.136, *p* < 0.001). Before treatment, the position was, on average, +2.5 mm and +2.3 mm for the right and left joints, respectively. After treatment, the position was, on average, −0.8 mm and −0.9 mm for the right and left joint, respectively (Table 2).

There was a statistically significant correlation between disc position in MRI and Th.P in CPV in 37 joints treated (Table 3) with the angular and coordinate method, r = 0.361, *p* = 0.028, *n* = 37.

Twenty-eight joints were treated with the angular method (75.6%) as the SCI was ≤70°. There was a statistically significant correlation between disc position in MRI and Th.P in CPV in joints treated with the angular method, r = 0.419, *p* = 0.027, *n* = 28 (Table 4)

Nine joints (24.3%) were treated with the coordinate method because the SCI was >70°.

There was no evidence of a positive correlation regarding nine joints with an angle of >70° SCI. For these joints, with an angle greater than 70 degrees, the association between MRI and Th.P was a negative correlation at −0.750, *p* = 0.020. This negative correlation in subjects with SCI was >70°. This could be explained by the therapeutic position, which had a greater Z value (vertical) than the X value (horizontal) (z > x). (Figure 4b illustrates that at a calculated Th.P of S 2.6 mm, the SCI is 85.5°. To obtain these values, we followed the condylographic tracing and the coordinates of the Th.P. were X = 0.21 and Z = 2.68. One can judge that the condylar displacement was more caudal than anterior, thus resulting in negative correlation.

### 3.2. Evaluation with Control Condylography

The (Figure 9) shows condylography before and after the CMR treatment.

The protrusion–retrusion in (Figure 9a) and open–close (Figure 9b) before the treatment show the clicking in both sides of the TMJ.

Once the joint luxations were initially reduced and after a 6-month follow-up of CMR therapy, a new condylography was performed. It was observed that the luxations did not re-appear as can be seen in (Figure 9c): protrusion–retrusion and in (Figure 9d): open–close. During control condylography, the movements begin from Th.P of the CMR stabilizers after localization of True Hinge Axis.

Table 5 shows the difference in the mean of excursive and incursive tracing in protrusion–retrusion and open–close before and after the CMR treatment.

Table 6 represents the significant values in the first, second, and third mm in protrusion–retrusion and open–close movements (n.s. = not significant).

The sensitivity of muscles and ligaments showed a significant improvement in pain on the VAS. The differences between the sensitivity to palpation in the RP, ICP, and ThP positions are shown in Table 7. (* is significant)

## 4. Discussion

The scientific literature shows that the anterior protrusive position of the mandible changes the disc–condyle relationship and this position is widely used in intra-articular TMD treatment for fabricating anterior repositioning splints [17].

The computerized condylography system has been one of the most widely used jaw tracking systems and studies in the past have confirmed its accuracy and reliability in condylar movement recording [37,38,39]. Condylography offers high sensitivity, specificity, and accuracy for the diagnosis of TMJ disc displacements [30,40]. The electronic determination of the sagittal condylar inclination for basic movements (protrusion–retrusion and open–close) is reliable and can be applied in patients irrespective of the status of the dentition [31]. When calculating the therapeutic position for controlled mandibular repositioning stabilizer (splint), precise condylography, MRI, and thorough clinical examination and diagnosis are basic requirements [33].

All the CMR stabilizers were fabricated from the therapeutic position obtained by the CMR method. Controlled condylographies from the CMR stabilizer position were performed at the end of the CMR treatment. (Table 5) shows the mean difference between the incursive and excursive tracings. It was observed that before treatment, there was a large separation between excursive and incursive tracings due to the displacement of the articular discs and condyles. Most subjects presented excursive tracings that were more cranial than the incursive tracings, which were more caudal. After treatment, once the luxations of discs were reduced, there was a significant reduction in the mean values of the differences between excursions and incursions in the first 1 to 3 mm in protrusion–retrusion and open–close movements (Table 6). This observation was possible because once the luxations were reduced, the protrusion and open–close tracings were more cranial, i.e., closer to the articular eminence. This correlates to previous studies [17].

The anterior repositioning splint can also be used in pain-related TMDs [41,42,43]. In the present study, on re-evaluation after 6 months, the VAS scores demonstrated the high efficiency of the CMR stabilizer treatment in terms of improving clinical symptoms and significant reduction in sensitivity to muscle palpation in Th.P as compared to the reference position and intercuspal position.

In the literature, there is a disagreement over the degree of mandible protrusion required for an anterior repositioning splint [15]. Due to this disagreement, most of the time, an overcorrection is performed by going too far anteriorly to an edge-to-edge position in past studies [15,17]. But after gradual grinding of ARS, the initial results decreased from 100%, 82%, and 78.1% to 72.5%, 64%, and 50%, respectively [17,44,45]. Meanwhile, in the present study, there is an optimal amount of controlled mandibular repositioning performed in increments to avoid an unnecessary reduction in ThP position during follow-up appointments, resulting in walking back of the condyles phenomenon.

The Th.P calculated by the angular and coordinate CMR method allows a precise calculation for fabrication of CMR stabilizers. In the present study, the MRI analysis shows a strong correlation between Th.P and the difference between the posterior band of the disc and the superior part of the condyle This strong correlation is due to the Th.P coming from the angular and coordinate method of CMR. There was a positive correlation in 28 joints where SCI was <70°. There was negative correlation in nine joints where the SCI > 70°, because when the eminence is steep, Th.P. is more caudal and less anterior (z > x).

Condylography is a valuable tool for precise calculation of sagittal condylar inclination. The average condylar inclination in healthy individuals varies from 42.1° ± 7.07° as compared to 33.25° when other methods are used [20].

In the present study, 24.3% of the study subjects had an SCI greater than 70°, which is far greater than the average SCI used to program an articulator. As described before, discrepancy in this sagittal condylar inclination may result in unnecessary adjustments on occlusion [19].

The first limitations of this pilot case–control study include the technique sensitivity of the CMR method and, secondly, the fact that a small number of patients were recruited and the fact that the follow-up lasted only 6 months. A future study with a larger number of participants is required, with a longer follow-up.

A future research direction will be the analysis of transversal displacement of the articulator disc before and after the implementation of the CMR method. Another future perspective to this study will be detailed evaluation of condylography data before and after the CMR therapy.

## 5. Conclusions

The CMR method was efficient, achieving up to a 97.3% reduction in T.M.J. joint luxations in subjects with DDwR. The results have verified that the therapeutic position, based on condylograhic data, calculated by the angular or coordinate method during controlled mandibular repositioning, was highly efficient.

What differentiates the CMR method from other methods is that the result of muscle palpation (VAS) is also taken into account in the search for an optimal therapeutic position where the muscles and ligaments are the least sensitive to palpation.

Also, the CMR method allows unwanted overcorrection.

In our study, by reducing the joint luxations, we observed a significant reduction in patients’ symptoms.

## Figures and Tables

**Figure 1 bioengineering-12-00865-f001:**
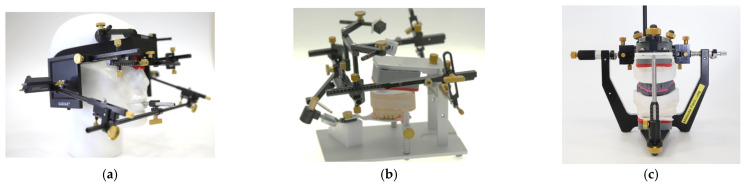
CADIAX 4 and Condylar Position Variator (CPV). (**a**) Kinematic facebow mounted on dental manikin; (**b**) mounted upper model on kinematic facebow; (**c**) mounted models on CPV. CADIAX 4, kinematic facebow and CPV: Gamma-dental, Klosterneuburg (GAMMA Medizinisch-wissenschaftliche Fortbildungs-GmbH) Klosterneuburg, Austria.

**Figure 2 bioengineering-12-00865-f002:**
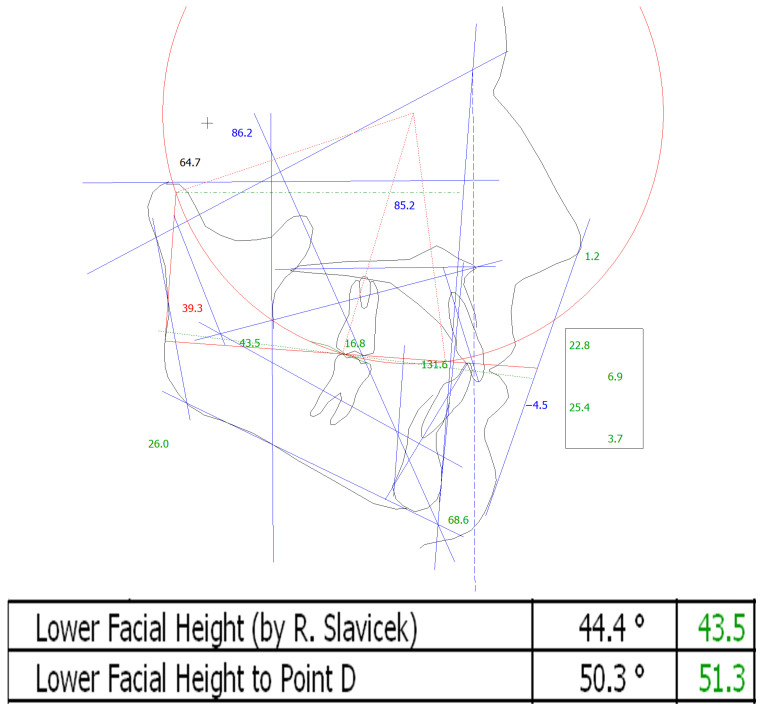
Cephalometric analysis.

**Figure 3 bioengineering-12-00865-f003:**
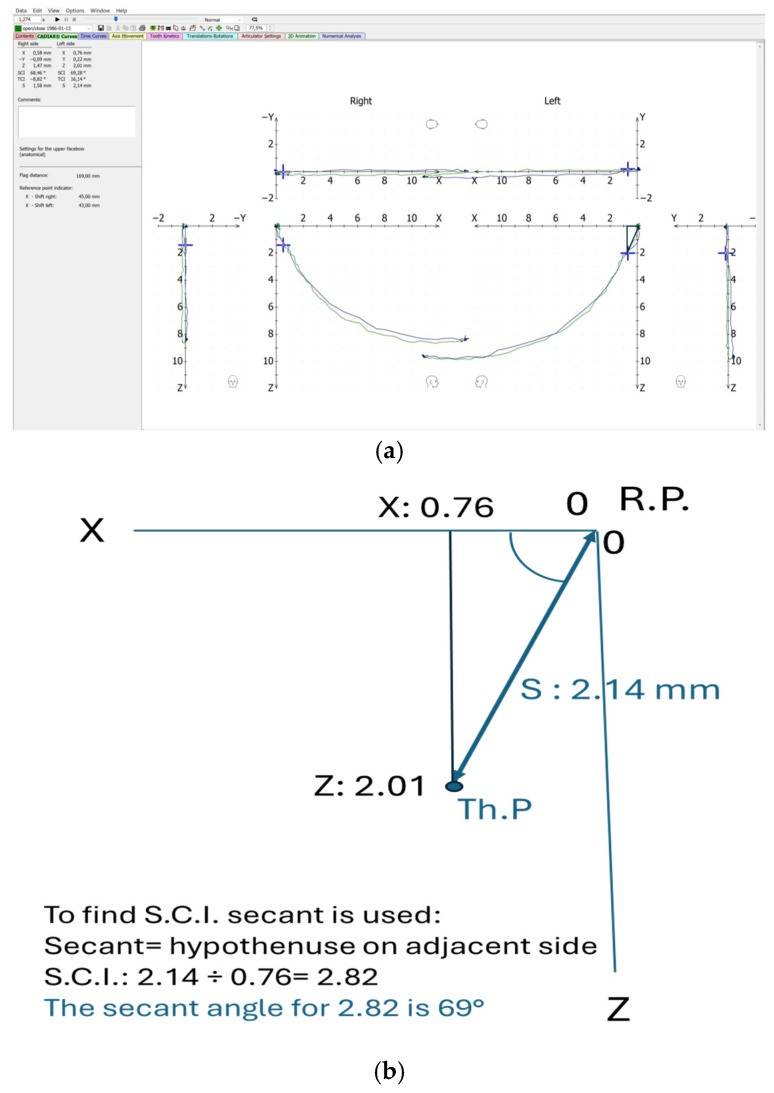
Condylography and angular method. (**a**) Condylography tracing showing left side Th.P; (**b**) calculation for angular method.

**Figure 4 bioengineering-12-00865-f004:**
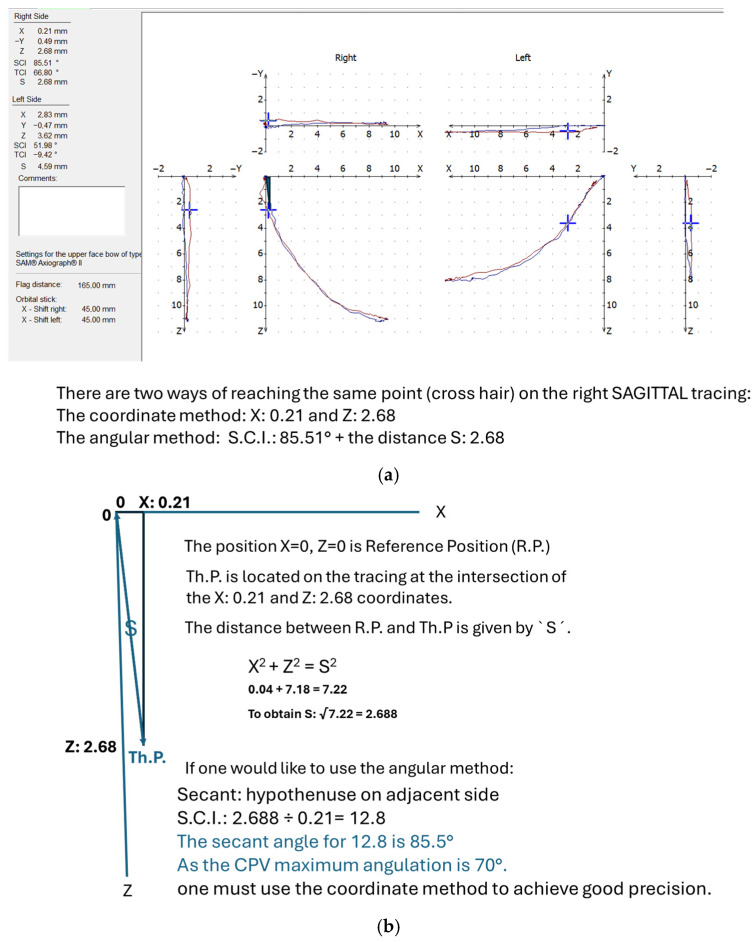
Condylography and coordinate method. (**a**) How to calculate Th.P on the right side; (**b**) calculation for the coordinate method.

**Figure 5 bioengineering-12-00865-f005:**
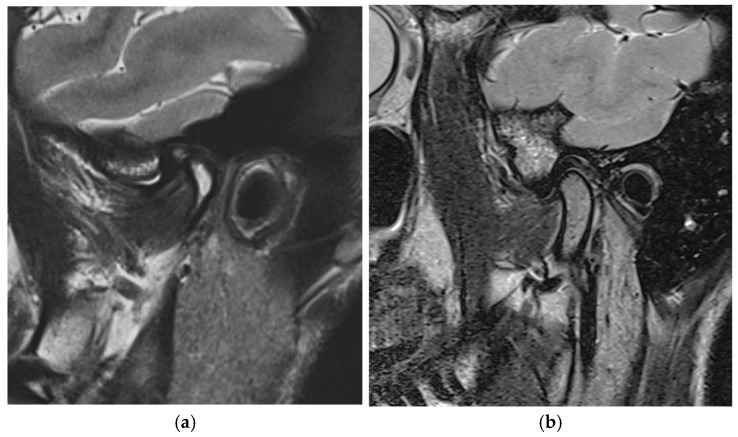
MRI before treatment and after CMR therapy. (**a**) Initial MRI; (**b**) MRI in Thp.

**Figure 6 bioengineering-12-00865-f006:**
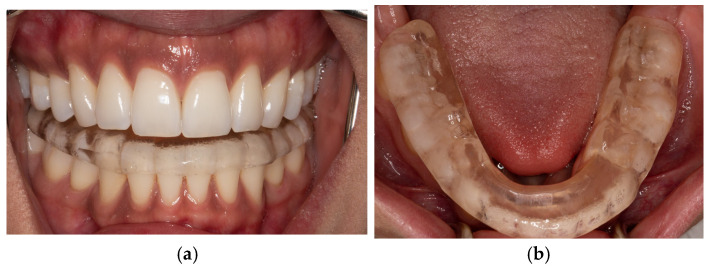
Controlled mandibular repositioning stabilizer (CMR stabilizer/splint). (**a**) Anterior view of CMR stabilizer. (**b**) Occlusal view of CMR stabilizer.

**Figure 7 bioengineering-12-00865-f007:**
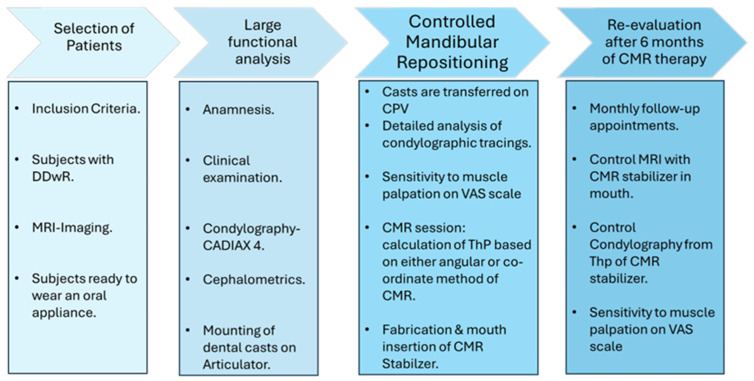
Diagrammatic representation of steps in the CMR method.

**Figure 8 bioengineering-12-00865-f008:**
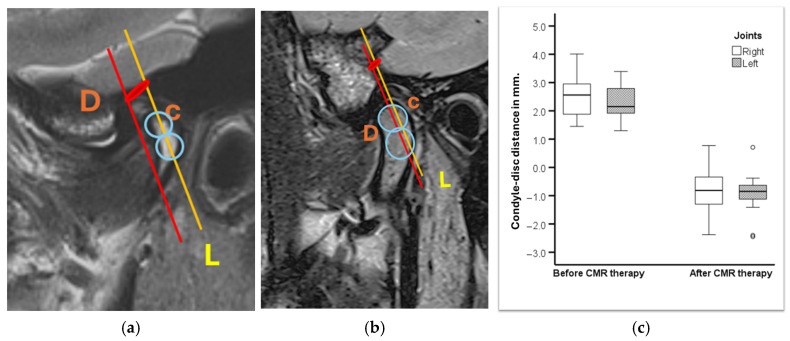
(**a**) Initial MRI; (**b**) MRI in THp; (**c**) measurement of C-D.

**Figure 9 bioengineering-12-00865-f009:**
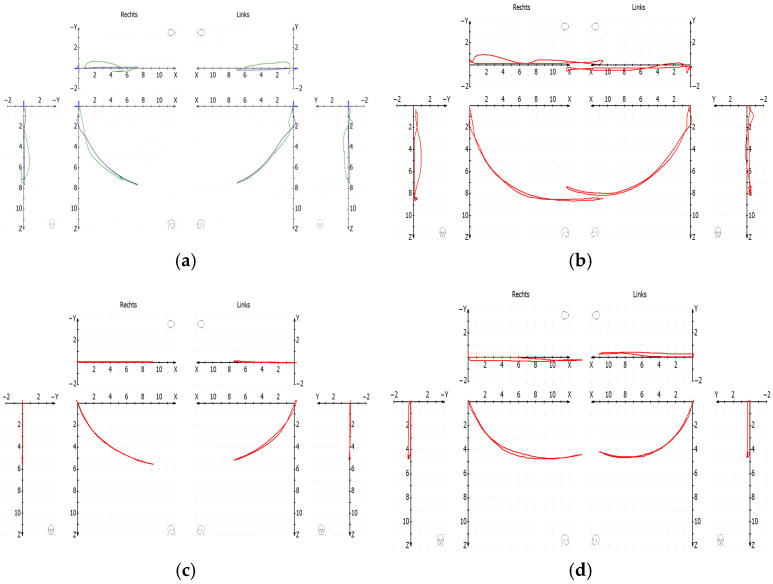
Sample condylography of one of the subjects before CMR therapy. (**a**) Protrusion–retrusion before treatment; (**b**) open–close before treatment; (**c**) protrusion–retrusion after CMR treatment; (**d**) open–close after CMR treatment.

**Table 1 bioengineering-12-00865-t001:** Patient distribution.

No. Patients	Joints with ADDwR	Females	Males	Age in Years Mean ± SD Range	Excluded Joints
20	37	18	2	28.7 ± 6.4 (18–41)	3 *

* These joints presented ADDw/oR (disc displacement without reduction) [33].

**Table 2 bioengineering-12-00865-t002:** Disc displacement in mm before and after CMR therapy.

	Before		After		*t*	*p*
	Mean ± SD	Min–Max	Mean ± SD	Min–Max		
Right (*n* = 18)	2.50 ± 0.67	1.45–4.01	−0.82 ± 0.71	−2.38–0.77	−13.533	<0.001
Left (*n* = 19)	2.33 ± 0.58	1.30–3.39	−0.94 ± 0.70	−2.45–0.71	−16.136	<0.001

Note: paired *t*-test.

**Table 3 bioengineering-12-00865-t003:** Correlation of Th.P to the angular and coordinate method of CMR.

Correlation		r	*p*	*n*
MRI with Th.P in CPV (s)	all Joints	0.361	0.028	37
MRI with Th.P in CPV (s)	<70°	0.419	0.027	28
MRI with Th.P (z) in CPV	>70°	−0.732	0.025	9
MRI with Th.P (x) in CPV	>70°	−0.573	0.107	9

Note: Spearman rank correlation: (There was a statistically significant correlation between disc position in MRI and Th.P in CPV in joints treated with the angular method, r = 0.361, *p* = 0.028, *n* = 37) MRI = difference before and after treatment in mm; Th.P = therapeutic position.

**Table 4 bioengineering-12-00865-t004:** Distribution of subjects treated by angular method and coordinate method.

MRI with Th.P CPV (s) (*n* = 37)	Percentage %	r	*p*	*n*
SCI < 70° (angular method)	75.6%	0.419	0.027	28
SCI > 70° (coordinate method)	24.3%	−0.750	0.020	9

Note: r = Spearman rank correlation; MRI = difference before and after treatment in mm; SCI = sagittal condylar inclination; thp = therapeutic position on x, z or $s=\sqrt{x^{2}+z^{2}}$ in mm.

**Table 5 bioengineering-12-00865-t005:** Distribution of difference in SCI in protrusion–retrusion (P/R) and open–close (O/C) in excursive versus incursive movements.

Movements		Difference Between Excursive and Incursive Tracings Before CMR Treatment								Difference Between Excursive and Incursive Tracings After CMR Treatment							
	Units:	1 mm	2 mm	3 mm	4 mm	5 mm	6 mm	7 mm	8 mm	1 mm	2 mm	3 mm	4 mm	5 mm	6 mm	7 mm	8 mm
P/R Left	MEAN	15.4	9.1	5.6	1.8	0.14	0.38	−0.4	−0.60	3.98	4.6	2.32	1.6	1.24	0.42	0.38	0.37
	STD	11.40	7.83	8.6	3.2	3.03	2.7	2.82	2.873	13.7	3.48	2.95	2.06	1.89	1.35	1.59	1.44
P/R Right	MEAN	15.84	7.3	1.2	1.9	1.6	0.95	0.54	0.278	7.1	3.04	1.98	1.49	0.96	0.67	1.19	1.44
	STD	11.6	6.3	7.6	3.2	2.3	2.05	1.64	0.946	4.46	2.50	2.56	2.09	1.58	1.59	1.71	1.88
O/C left	MEAN	12.06	7.2	3.1	1.7	0.4	−0.25	−0.50	−0.24	15.54	17.75	−0.10	−0.37	−0.55	−1.7	−0.09	−0.52
	STD	10.8	5.9	4.5	4.0	3.4	3.1	2.7	2.61	53.5	55.40	3.2	3.1	2.83	3.61	2.25	2.86
O/C right	MEAN	5.84	8.7	3.5	1.2	−0.35	−0.6	−0.85	−0.56	3.39	1.2	0.16	0	−0.10	0.32	0.13	−0.27
	STD	40.4	8.5	6.8	5.5	4.69	3.4	2.99	2.3	8.65	4.3	2.82	2.4	2.15	1.79	1.76	1.3

**Table 6 bioengineering-12-00865-t006:** Difference in SCI in protrusion–retrusion and open–close in excursive versus incursive movements.

*N* = 20 Subjects	Protrusion–Retrusion Left	Protrusion–Retrusion Right	Open–close Left	Open–close Right
mm	Significance	Significance	Significance	Significance
1 mm	* <0.01	* <0.01	* <0.01	n.s.
2 mm	n.s.	* <0.05	* <0.01	* <0.01
3 mm	n.s.	n.s.	* <0.01	* <0.05
4 mm	n.s.	n.s.	* <0.05	n.s.
5 mm	n.s.	n.s.	n.s.	n.s.
6 mm	n.s.	n.s.	* <0.05	n.s.
7 mm	n.s.	n.s.	n.s.	n.s.
8 mm	n.s.	n.s.	n.s.	n.s.
9 mm	n.s.	n.s.	n.s.	n.s.
10 mm	n.s.	n.s.	n.s.	n.s.

* Significant.

**Table 7 bioengineering-12-00865-t007:** Mean and standard deviation of sensitivity to muscle palpation in different positions on the left (*n* = 19) and the right side (*n* = 18) on VAS (0–10).

Muscles	Left			Right		
	RP	ICP	CMR	RP	ICP	CMR
TubMaxillae	3.5 ± 2.3	3.3 ± 2.2	0.1 ± 0.2	4.1 ± 2.4	4.3 ± 3.0	0.1 ± 0.3
MedPterygoid	3.4 ± 3.0	3.5 ± 3.3	0.3 ± 0.5	3.1 ± 2.9	3.4 ± 2.9	0 ± 0
Mylohyoid	0.8 ± 1.7	0.5 ± 1.1	0 ± 0	0.4 ± 0.8	0.3 ± 0.7	0 ± 0
SupMasseter	2.1 ± 2.2	2.2 ± 1.9	0.1 ± 0.5	0.8 ± 1.0	1.5 ± 2	0 ± 0
TempTendon	2.2 ± 2.0	2.4 ± 2.1	0 ± 0	1.5 ± 1.3	2.2 ± 2.2	0 ± 0
Digastric	0.6 ± 1.0	0.7 ± 1.2	0 ± 0	0.4 ± 0.8	1.2 ± 1.6	0 ± 0
Lateral Pole	0.3 ± 0.8	0.6 ± 1.2	0 ± 0	0.9 ± 1.7	0.6 ± 0.9	0 ± 0
DeepMasseter	1.3 ± 1.5	1.7 ± 1.8	0.1 ± 0.3	1.3 ± 1.4	1.6 ± 1.5	0.1 ± 0.3
TMLigament	1.5 ± 1.6	2.2 ± 2.1	0.2 ± 0.6	1.8 ± 1.8	2.1 ± 2	0.1 ± 0.5
RJSpace	1.4 ± 1.3	1.0 ± 1.6	0.1 ± 0.5	1.1 ± 1.4	1.1 ± 1.7	0 ± 0
OmoHyoid	0.9 ± 1.5	1.2 ± 1.9	0.3 ± 0.7	1.0 ± 1.6	0.7 ± 1.5	0.3 ± 0.7
U.Trapezius	1.4 ± 2.0	2.3 ± 2.4	0.2 ± 0.5	1.8 ± 2.3	2.3 ± 2.4	0.2 ± 0.5
PostTemporalis	0.8 ± 1.7	1.1 ± 1.5	0.1 ± 0.5	1.0 ± 1.4	1.0 ± 1.6	0.1 ± 0.5

Note: (RP = reference position; ICP = intercuspal position; CMR/Th.P = therapeutic position).

## Data Availability

Data is available upon request.

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
