# Peer review of "Controlled Mandibular Repositioning: A Novel Approach for Treatment of TMDs"

_bioengineering, 2025, doi:10.3390/bioengineering12080865_

Round 1

Reviewer 1 Report

Comments and Suggestions for Authors

This paper is an interesting study using a CMR stabilizer and is well written.

  1. As for the vertical dimension of occlusion (V.D.O.) evaluated from the digitized cephaometric X-Ray, it would be easier to understand if you explained it with a diagram.
  2. The TMJ disc in Fig. 4a and Fig. 5a does not appear to be anteriorly displaced. Can you provide a diagram showing a more typical TMJ disc displacement?
  3. Furthermore, can you provide intraoral photographs with a CMR stabilizer attached? This would make for a more persuasive paper.

Author Response

Dear Professor Reviewer 1, Thanks for guiding and suggesting the needful changes as it has improved our manuscript, and we have done all the changes,   Thanking you. 

Comment 1:  As for the vertical dimension of occlusion (V.D.O.) evaluated from the digitized cephaometric X-Ray, it would be easier to understand if you explained it with a diagram.

 Response 1: As advised by Professor Reviewer1 a new fig 2 is added with cephalometric analysis showing lower facial height. From line 212 to 215 there is a explanation for vertical dimension of occlusion.

Comment 2:  The TMJ disc in Fig. 4a and Fig. 5a does not appear to be anteriorly displaced. Can you provide a diagram showing a more typical TMJ disc displacement?

Response 2: As advised by Professor Reviewer1 a new image has been replaced in fig 5a showing initial MRI with disc displaced anteriorly and condyle displaced posteriorly.

Comment 3:  Furthermore, can you provide intraoral photographs with a CMR stabilizer attached? This would make for a more persuasive paper.

Response 3: As advised by Professor Reviewer1 a new fig 6 (Fig 6a and fig6 b) are inserted in line 258 showing CMR stabilzer .

Reviewer 2 Report

Comments and Suggestions for Authors

This paper offers a thorough and technical analysis of a new approach to treating temporomandibular joint disorders (TMDs), particularly disc displacement with reduction (DDwR), called controlled mandibular repositioning (CMR). The paper is well-organized and informative; however, it would benefit from addressing a few areas.

  • The authors mention the novelty of the CMR method but provide limited comparative discussion with other established methods beyond ARS and splint therapy. A more robust contrast with conventional protocols would better highlight the innovation. Can you elaborate on how CMR quantitatively improves upon ARS therapy in terms of long-term joint health, patient-reported outcomes, or relapse rates?
  • The figure legends could be more descriptive, for example, in figure 5, it would be beneficial to clearly describe what (a), (b), and (c) indicate.
  • The authors might consider adding a figure or flow diagram early in the methodology to show how these measurements and techniques are interrelated during the treatment planning. Certain abbreviations like SCI, RP, CPV are used extensively and having a flowchart will make it easier for the readers.
  • Table 3 reports both the positive and negative correlation between MRI outcomes and coordinate/angular methods. It would be beneficial to develop on its interpretation which is currently very brief. Could the inverse relationship for SCI>70 be due to anatomical or neuromuscular compensation?

Author Response

Dear Professor Reviewer, Thanks for highlighting and suggesting the valuable corrections for improving our Manuscript, Thanking you.

Comment 1:  The authors mention the novelty of the CMR method but provide limited comparative discussion with other established methods beyond ARS and splint therapy. A more robust contrast with conventional protocols would better highlight the innovation. Can you elaborate on how CMR quantitatively improves upon ARS therapy in terms of long-term joint health, patient-reported outcomes, or relapse rates?

Response 1: As the study was designed to see the optimal amount of repositioning required to recapture the displaced disc where ARS was most commonly used in DDwR and in our study subjects we had only included patients with DDwR so thats why we did not included other types of splints therapy. The study has an ethical approval till 2026 and it is planned for long term follow up with new condylography and new control MRI and may be in future a long term follow up can be provided.  This long term control joint health is one of the  limitation  we have also mentioned this point in  our study.

Comment 2: The figure legends could be more descriptive, for example, in figure 5, it would be beneficial to clearly describe what (a), (b), and (c) indicate.

Response 2: For this part we have made the changes  for the figure 6 which was initially fig 5. and in line 299 till 306 we have made the changes and added legends as suggested by you for the better clarity which was not before. 

Comment 3:The authors might consider adding a figure or flow diagram early in the methodology to show how these measurements and techniques are interrelated during the treatment planning. Certain abbreviations like SCI, RP, CPV are used extensively and having a flowchart will make it easier for the readers.

Response 3: For this part we have added a new diagrammatic flowchart  on CMR method explaining the series of steps done (line 265) . and extended abbreviations list in (line 483) onwards

Comment 4: Table 3 reports both the positive and negative correlation between MRI outcomes and coordinate/angular methods. It would be beneficial to develop on its interpretation which is currently very brief. Could the inverse relationship for SCI>70 be due to anatomical or neuromuscular compensation?

Response 4: Dear reviewer thanks for highlighting this very important information and with help of our team we have tried to explain it and yes it is due to anatomical compensation and we have tried to explain it in line 339-line 343 in manuscript which is also illustrated in fig 4b of co-ordinate method.

Reviewer 3 Report

Comments and Suggestions for Authors

This study introduces the Controlled Mandibular Repositioning (CMR) method, which integrates clinical examination, imaging (MRI), computerized cephalometry, computerized condylography, neuromuscular palpation and Condylar Position Variator (CPV) to define an individualized therapeutic position (Th.P). This study demonstrates the effectiveness of the CMR method to find a precise therapeutic position resulting in 97.3% of joint reduction in disc displacement with reduction (DDwR). It is underscored the importance of precise, individualized Th.P determination for effective anterior repositioning.
Very significant shortcomings and omission of the paper are the following:
1. All references in the text should be listed sequentially according to their appearance and enclosed in square brackets.
2. All abbreviations must be deciphered at first appearance in text. For Example: OPGs and CBCT (Line 89), CADIAX (Line 107)
3. (Line 133): “t-test” must be accompanied by corresponding link and brief description.
4. (Line 248): “Shapiro Wilk test” must be accompanied by corresponding link and brief description.
5. (Line 252 – 253): “Wilcoxon signed-rank test” must be accompanied by corresponding link and brief description.
6. (Line 273): “Spearman rank-correlation” must be accompanied by corresponding link and brief description.
7. All MDPI rules for figure captions have been violated.
8. All text of the paper is fragmentary (see Line 299) and noncompleted without necessary explanations. Tables 3-7 are discussed in Section 4. Discussion, away from their presence in the text.
9. Table 5: What is the “P/R left” and other designations? They differ in the same designations in Table 6.
10. Table 6: What is the n.s.?
11. Conclusions are absent practically.
12. All references in the list of references should be accompanied by doi.

Author Response

Comment1. All references in the text should be listed sequentially according to their appearance and enclosed in square brackets.

Response: This part has been corrected and now we have square brackets

Comment2. All abbreviations must be deciphered at first appearance in text. For Example: OPGs and CBCT (Line 89), CADIAX (Line 107)

Response: This correction is done.

Comment3. (Line 133): “t-test” must be accompanied by corresponding link and brief description.

Response: This correction is done .

Comment4. (Line 248): “Shapiro Wilk test” must be accompanied by corresponding link and brief description.

Response: This correction is done.

Comment5. (Line 252 – 253): “Wilcoxon signed-rank test” must be accompanied by corresponding link and brief description.

Response: This correction is done.

Comment6. (Line 273): “Spearman rank-correlation” must be accompanied by corresponding link and brief description.

Response: This correction is done.

Comment7. All MDPI rules for figure captions have been violated.

Response: We have tried to do this part.

Comment8. All text of the paper is fragmentary (see Line 299) and noncompleted without necessary explanations. Tables 3-7 are discussed in Section 4. Discussion, away from their presence in the text.

Response: This correction is done and extras has been removed.

Comment9. Table 5: What is the “P/R left” and other designations? They differ in the same designations in Table 6.

Response: We have corrected it and written complete Protrusion-retrusion.

Comment10. Table 6: What is the n.s.?

Response: It was not initially mentioned now we have written (not significant)

Comment11. Conclusions are absent practically.

Response: We have worked on it and should be ok.

Comment12. All references in the list of references should be accompanied by doi.

Response: Yes the doi was not there but in new version doi has been added.

Round 2

Reviewer 3 Report

Comments and Suggestions for Authors

The authors have performed significant work on improving the manuscript. However, some questions remain:

  1. MDPI rules for figure captions have been violated.
  2. Conclusions are absent practically.
  3. All references in the list of references must be formed according to the MDPI rules for authors.

Author Response

Comment 1: MDPI rules for figure captions have been violated.

Resposne1: Dear Reviewer as per your suggestion i have corrected all the Figure heading in format MDPI_2.2_heading 2 as given by template and also the table heading to MDPI_4.1_table_caption.

Comment 2: Conclusions are absent practically.

Response 2:  This part, as per your suggestions  is also corrected and we have mentioned the CMR method allows unwanted overcorrections, which is the key point of this study and is very effective in treating disc displacement with reduction type of temporomandibular joint disorders. We have also mentioned in discussion part as in literature, few of authors were bringing the lower jaw to an edge to edge position i.e. doing overcorrections and then during follow-up appointments subsequent grinding of the anterior repositioning splint was done in  order to reach the therapeutic position, on the contrary in our study with CMR method based on patients individual condylar movements right from beginning a precise therapeutic position cal be calculated in combination with muscle palpation on VAS scale. 

Comment 3: All references in the list of references must be formed according to the MDPI rules for authors.

Response 3: Dear Reviewer as suggested in reference list the references has been induced as mentioned in template from bioengineering ( Author 1, A.B.; Author 2, C.D. Title of the article. Abbreviated Journal Name Year, Volume, page range.+ DOI number).   Dear Reviewer, all the changes in 3rd version has been highlighted in yellow.

Dear Reviewer i just also wanted to add here after email from Mr Ye Rod,

He mentioned (Regarding points 1 and 3, please note that the manuscript will undergo full formatting and layout by our internal editors upon acceptance).

 I have still worked on all the 3 points  after your suggestions and has been corrected and  edited.

Hope its all fine if, something is still there for corrections please guide us.

Kind Regards

Diwakar Singh